**Data Availability Statement:** The data underlying the results presented in the study are available from the U.S. Center for Disease Control and

# Association between alanine aminotransferase within the normal range and all-cause and cause-specific mortality: A nationwide cohort study

**Aayush Visaria** *, **Suraj Pai, Alla Fayngersh, Neil Kothari**

Department of Medicine, Rutgers New Jersey Medical School, Newark, New Jersey, United States of America

* aayush.visaria@rutgers.edu

## Abstract

### Background and aim

We sought to determine the association between alanine aminotransferase (ALT) in the normal range and mortality in the absence of liver dysfunction to better understand ALT's clinical significance beyond liver injury and inflammation.

### Methods

A cohort of 2,708 male and 3,461 female adults aged 20–75 years without liver dysfunction (ALT<30 in males & <19 in females, negative viral serologies, negative ultrasound-based steatosis, no excess alcohol consumption) from the National Health and Nutrition Examination Survey (NHANES)-III (1988–1994) were linked to the National Death Index through December 31, 2015. Serum ALT levels were categorized into sex-specific quartiles (Females: <9, 9–11, 11–14, ≥14 IU/L, Male: <12, 12–15, 15–20, ≥20 U/L). The primary outcome was all-cause mortality. Hazard ratios (HRs) were estimated, adjusting for covariates and accounting for the complex survey design.

### Results

Relative to males in the lowest quartile (Q1), males in the highest quartile (Q4) had 44% decreased risk of all-cause mortality (aHR [95% CI]: 0.56 [0.42, 0.74]). Females in Q4 had 45% decreased risk of all-cause mortality (aHR [95% CI]: 0.55 [0.40, 0.77]). Males with BMI <25 kg/m² in Q4 had significantly lower risk of all-cause mortality than Q1; however, this association did not exist in males with BMI ≥25 (BMI<25: 0.36 [0.20, 0.64], BMI≥25: 0.77 [0.49, 1.22]). Risk of all-cause mortality was lower in males ≥50 years than in males<50 (age≥50: 0.55 [0.39, 0.77], age<50: 0.81 [0.39, 1.69]). These age- and BMI-related differences were not seen in females.

### Conclusion

ALT within the normal range was inversely associated with all-cause mortality in U.S. adults.

Prevention at https://wwwn.cdc.gov/nchs/nhanes/nhanes3/datafiles.aspx#core. This website is the direct link to the NHANES-III (1988-1994) data files, available for download in DAT format along with sample SAS codes.

**Funding:** The author(s) received no specific funding for this work.

**Competing interests:** The authors have declared that no competing interests exist.

## Introduction

Serum alanine aminotransferase (ALT), among other enzymes, is a marker of liver function. In the liver, it is found in the cytosol, where it catalyzes the metabolism of amino acids [1, 2]. Changes in aminotransferase levels can be pathologic; elevations may signify direct injury or inflammation whereas reductions may reflect diminished synthetic ability. Both are instances of hepatocellular dysfunction albeit with differing laboratory findings [1].

Despite ALT's predominantly hepatocellular origin, it is also found in various other cells, including skeletal muscles [3]. Studies have linked its activity with all-cause mortality; however, evidence is inconsistent regarding the direction and magnitude of the association, in part due to differences in ALT ranges included, incorporation of confounders, and probable differences in study populations [3–8].

Because of ALT's multiple functions, we sought to determine the association between increasing ALT within the normal range [4–6] and long-term mortality in U.S. adults without liver dysfunction, in an attempt to exclude ALT's activity in liver injury/inflammation. Few studies, if any, to our knowledge have rigorously restricted their study populations to normal ALT levels, excluded underlying liver dysfunction, and accounted for ALT's age- and sex-related differences in distribution [6–11]. By incorporating these criteria into this population-based cohort study, our purpose is to further describe ALT's potentially independent role in long-term all-cause and cause-specific mortality, stratified by sex, age group, and BMI.

## Methods

### NHANES dataset and linkage

The National Health and Nutrition Examination Survey (NHANES)-III (1988–1994) is a nationally representative survey of the civilian, non-institutionalized United States population. The survey, implemented by the National Center for Health Statistics (NCHS), follows a complex, stratified, multistage probability design consisting of interviews, physical examinations at home or at a mobile examination center, and laboratory testing. Further information about the dataset and the sample design can be found elsewhere [12]. Data from NHANES-III is also linked to death certificates from the National Death Index (NDI) as of December 31st, 2015, allowing for mortality analysis. This study was deemed exempt by Rutgers' Institutional Review Board because the dataset is publicly available and completely deidentified.

### Study population

A total of 2,708 male and 3,461 non-pregnant female individuals aged 20–75 years were included in this study. Initially, the NHANES III population consisted of 33,994 individuals, 18,825 (8,184 males and 10,009 females) of whom were adults ≥20 years and subsequently assessed for study eligibility. We included individuals with data on serum ALT levels, mortality status, and covariates determined a priori. To rigorously exclude individuals with any liver dysfunction, we used similar criteria as established in prior population-based studies [4, 5, 13] and also excluded those with hepatic steatosis as per ultrasound. We excluded those with ALT >19 IU/L in females (N = 1,193) and ALT >30 IU/L in males (N = 982) as per the 2002 definition of healthy-range ALT [6]. We also excluded those with evidence of moderate-to-severe hepatic steatosis on ultrasound readings (N = 3,147 males and 3,280 females), excessive alcohol consumption (>10 drinks/week for females [N = 507] and >20 drinks/week for males [N = 596]), iron overload (transferrin saturation >50%, N = 164 males and 112 females), and positive hepatitis B or C serology (N = 281 males and 305 females). Finally, we excluded those with unknown values for covariates (N = 306 males and 519 females).

## Variables of interest

Our primary predictor was serum ALT levels. Serum ALT levels were measured using a Hitachi Model 737 multichannel analyzer (Boehringer Mannheim Diagnostics, Indianapolis, IN). Briefly, a kinetic assay was performed to determine the rate of consumption of nicotinamide adenine dinucleotide (NADH) in the reaction of alpha-ketoglutarate and L-alanine to form L-glutamate and pyruvate [14]. ALT levels were recorded to the nearest 1 IU/L. ALT was then categorized into sex-specific quartiles (Females: <9, 9–11, 11–14, ≥14 IU/L, Male: <12, 12–15, 15–20, ≥20 IU/L).

Our primary outcome was all-cause mortality through December 31$^{st}$, 2015, as determined by linkage with the NDI. Secondary outcomes included 10-year all-cause mortality, cardiovascular mortality, and cancer mortality. Cause of death was categorized into 9 leading causes of death using a unified system spanning ICD-9 and ICD-10 codes (variable name: UCOD_LEADING). A cardiovascular cause of death was defined as a death from heart disease (ICD-10: I00-I09, I11, I13, I20-I51) or cerebrovascular disease (ICD: I60-I69). A cancer cause of death was defined by ICD-10 codes (or ICD-9 equivalents) of C00-C97 [15].

We included the following covariates in our analysis: demographic variables (age, race/ethnicity [Non-Hispanic White, Non-Hispanic Black, Mexican American, Other], poverty-income ratio [median household income / poverty threshold]), sociobehavioral variables (alcohol consumption, smoking status [ever smoker, never smoker], total caloric intake), cardiometabolic variables (waist circumference, HDL, systolic blood pressure, triglycerides, Homeostatic Model Assessment of Insulin Resistance [HOMA-IR], C-reactive protein, presence of albuminuria [urine albumin-creatinine ratio >30], self-reported history of cardiovascular disease), and liver function-related variables (albumin, platelet count, AST, hemoglobin level, total bilirubin). Measurement protocols for each covariate are described elsewhere [12]. Ultrasound readings to exclude those with moderate-to-severe hepatic steatosis as described above were determined through revisited gallbladder ultrasounds.

## Statistical analysis

To compare baseline characteristics across ALT quartiles, we used ANOVA for continuous variables and Rao-Scott Chi-square tests for categorical variables, followed by post hoc pairwise comparisons between quartiles using Tukey's multiple comparison test.

To determine the association between serum ALT quartiles and mortality, we used Cox proportional hazard models (proportionality assumption verified using Schoenfeld residual plots) for all-cause mortality and Fine & Gray competing risks models for cause-specific mortality, adjusting for covariates in a series of 4 models. Model 1 was unadjusted for covariates. Model 2 adjusted for demographic and sociobehavioral factors. Model 3 adjusted for Model 2 covariates + cardiometabolic factors. Model 4 adjusted for Model 3 covariates + liver function-related factors. Analysis was additionally stratified a priori by age (<50, ≥50) and BMI (≥25, <25) to assess hypotheses of ALT's association with mortality. We conducted a sensitivity analysis treating ALT as a continuous variable. To better speculate on ALT's age-related changes, we plotted mean baseline liver enzyme levels across 5-year age groups. All analyses utilized weights to produce nationally representative estimates and account for the oversampling of minorities (e.g. elderly, African Americans, and Mexican Americans). We did not adjust statistical significance for multiple comparisons, as post-hoc testing and stratified analyses were considered exploratory in nature. All analyses also accounted for the complex survey design with strata and cluster variables using a traditional Taylor linearization approach to obtain accurate variance estimates. Analyses were conducted using SAS version 9.4 (SAS Institute Inc., Cary, NC) with a significance level of 0.05.

## Results

### Baseline characteristics by ALT quartiles

There was a total of 2,708 male and 3,461 female adults included in the study. The average age (SD) among males was 41.5 (0.5) years and 41.7 (0.5) years in females (see Table 1 and S1 Table). The mean (SD) ALT levels in males and females were 16.4 (0.3) and 11.3 (0.2) U/L, respectively. Males had higher average levels of gamma glutamyl transferase (GGT), albumin, and total bilirubin than females.

Among males, those in the highest quartile of ALT (Q4, ≥20 IU/L) were younger, less likely to have hypertension, albuminuria, and insulin resistance, and had lower proportions of self-reported cardiovascular complications (MI, stroke, CHF) compared to those in the lowest quartile (Q1, <12 IU/L); however, males in Q4 had higher BMI, lower HDL, and higher tri-glyceride levels compared to males in Q1. There was no significant difference in race/ethnicity, smoking status, alcohol consumption, or macronutrient caloric intake among the quartiles. As expected, males in Q4 had higher GGT, aspartate aminotransferase (AST), lactate dehydrogenase (LDH), and hemoglobin levels compared to those in Q1. On the contrary, there was no significant difference in total bilirubin levels, albumin levels, or platelet count across quartiles (Table 1).

Females have similar patterns of baseline characteristics across quartiles, although there are some clear differences in demographic, sociobehavioral, and comorbidity factors (S1 Table). Females in Q4 (≥14 IU/L) were older, less likely to be non-Hispanic Black, less likely to have smoked, and had lower average alcohol consumption compared to females in Q1 (<9 IU/L). Females in Q4 also had a significantly greater proportion of diabetes, hypertension, metabolic syndrome, and insulin resistance.

### Modeling the association between ALT quartiles and all-cause mortality in males

Overall, 940 (33%) of males died from all causes through a median 22.8 years of follow-up. Among these deaths, 309 (11%) died within 10 years of follow-up, 249 (6.3%) died from car-diovascular causes, and 214 (6.1%) died from cancer.

There was a significant inverse association between ALT quartile and all-cause mortality, after adjusting for all covariates (see Table 2). Males in Q4 had 44% lower risk of all-cause mor-tality compared to males in Q1. When stratifying by BMI (≥25, <25 kg/m$^2$), we found a signif-icant 64% decreased adjusted risk of all-cause mortality for those in Q4 vs. Q1 for normal BMI participants (aHR [95% CI]: 0.36 [0.20, 0.64]), but no significant association in elevated BMI participants (aHR [95% CI]: 0.77 [0.49, 1.29], Table 3).

### Modeling the association between ALT quartiles and All-cause mortality in females

Among females, 861 (23%) died from all causes through a median 22.9 years of follow-up. Of these deaths, 243 (6.5%) died within 10 years of follow-up, 234 (6.2%) died from cardiovascular causes, and 220 (5.8%) died from cancer.

Just as in males, there was a significant inverse association between ALT quartile and all-cause mortality, after adjusting for all covariates (Table 4). Females in Q4 had 45% lower risk of all-cause mortality compared to females in Q1. In contrast to males, we found that, irrespec-tive of BMI category, females in Q4 had significantly lower risk of all-cause mortality com-pared to females in Q1 (Table 3).

**Table 1. Baseline characteristics by ALT quartile in males.**

| | Overall | Quartile 1 (<12 IU/L) | Quartile 2 (12–15 IU/L) | Quartile 3 (15–20 IU/L) | Quartile 4 (≥20 IU/L) | P-value* (1 vs. 2) | P-value* (1 vs. 3) | P-value* (1 vs. 4) | Overall# P-value |
|---|---|---|---|---|---|---|---|---|---|
| **Demographics** | | | | | | | | | |
| Age, Mean (SD) | 41.5 (0.5) | 44.7 (1.1) | 42.1 (0.9) | 40.9 (0.8) | 38.9 (0.6) | 0.051 | 0.0059 | <0.0001 | <0.0001 |
| Race/Ethnicity | | | | | | | | | |
| White | 1121 (79%) | 271 (78%) | 267 (81%) | 299 (78%) | 284 (79%) | | | | 0.2774 |
| Black | 737 (9%) | 212 (12%) | 156 (9%) | 185 (8%) | 184 (8%) | | | | |
| Mexican American | 755 (5%) | 124 (4%) | 162 (4%) | 238 (5%) | 231 (5%) | | | | |
| Other | 95 (7%) | 21 (6%) | 15 (6%) | 32 (9%) | 27 (8%) | | | | |
| Poverty-Income Ratio | 3.30 (0.08) | 3.09 (0.13) | 2.98 (0.09) | 3.52 (0.12) | 3.45 (014) | 0.39 | 0.016 | 0.098 | 0.0013 |
| **Sociobehavioral Factors** | | | | | | | | | |
| Smoking Status | 1654 (61%) | 428 (66%) | 371 (61%) | 441 (60%) | 414 (59%) | | | | 0.33 |
| Alcohol consumption | 5.1 (0.5) | 3.9 (0.5) | 5.8 (1.0) | 5.8 (0.7) | 4.6 (0.7) | 0.077 | 0.025 | 0.39 | 0.0676 |
| Exercise Status ("Jog/Run in the past month?") | 547 (21%) | 109 (21%) | 128 (22%) | 157 (18%) | 153 (24%) | | | | 0.401 |
| Carbohydrate Intake (kcal) | 1295 (20) | 1243 (44) | 1293 (43) | 1301 (35) | 1337 (35) | 0.42 | 0.28 | 0.1 | 0.38 |
| Protein Intake (kcal) | 402 (6) | 387 (12) | 396 (12) | 402 (11) | 418 (12) | 0.54 | 0.34 | 0.093 | 0.39 |
| Total fat intake (kcal) | 923 (16) | 906 (36) | 901 (33) | 931 (20) | 959 (28) | 0.91 | 0.59 | 0.2 | 0.4 |
| Total caloric intake (kcal) | 2615 (34) | 2526 (80) | 2593 (75) | 2636 (45) | 2707 (66) | 0.55 | 0.28 | 0.067 | 0.24 |
| **Examination Measurements** | | | | | | | | | |
| Heart Rate (beats-per-min) | 87.7 (4.0) | 92.5 (9.4) | 74.3 (1.9) | 88.5 (7.7) | 93.1 (8.3) | 0.0585 | 0.75 | 0.96 | 0.0029 |
| BMI (kg/m$^2$) | 25.7 (0.1) | 24.9 (0.2) | 25.0 (0.2) | 25.8 (0.2) | 26.6 (0.2) | 0.69 | 0.0019 | <0.0001 | <0.0001 |
| Waist Circumference (cm) | 92.7 (0.3) | 91.4 (0.7) | 90.8 (0.6) | 93.4 (0.4) | 94.3 (0.6) | 0.51 | 0.0083 | 0.0034 | 0.0001 |
| Thigh Circumference (cm) | 51.4 (0.1) | 50.0 (0.3) | 50.6 (0.2) | 51.8 (0.3) | 52.6 (0.3) | 0.045 | <0.0001 | <0.0001 | <0.0001 |
| Systolic Blood Pressure (mmHg) | 122.1 (0.5) | 123.5 (1.0) | 123.2 (0.8) | 121.4 (1.0) | 121.4 (0.7) | 0.84 | 0.14 | 0.07 | 0.1052 |
| Diastolic Blood Pressure (mmHg) | 75.4 (0.3) | 74.6 (0.6) | 74.7 (0.6) | 75.8 (0.6) | 76.2 (0.5) | 0.86 | 0.15 | 0.071 | 0.1326 |
| **Laboratory Measurements** | | | | | | | | | |
| Plasma Glucose (mg/dl) | 97.7 (1.0) | 96.4 (0.7) | 97.2 (1.3) | 98.9 (3.2) | 97.2 (1.0) | 0.58 | 0.45 | 0.51 | 0.77 |
| Hemoglobin A1c (%) | 5.33 (0.02) | 5.36 (0.03) | 5.29 (0.05) | 5.36 (0.06) | 5.30 (0.04) | 0.2 | 0.93 | 0.27 | 0.47 |
| Serum Insulin (uU/mL) | 9.3 (0.2) | 8.4 (0.3) | 8.2 (0.4) | 9.6 (0.7) | 10.4 (0.5) | 0.7 | 0.14 | 0.0008 | 0.0019 |
| Serum HDL (mg/dl) | 45.8 (0.5) | 45.9 (0.8) | 47.8 (0.7) | 45.8 (0.7) | 44.0 (0.8) | 0.0724 | 0.94 | 0.0799 | 0.0069 |
| Serum LDL (mg/dl) | 132 (2) | 133 (3) | 132 (2) | 128 (3) | 135 (3) | 0.67 | 0.31 | 0.78 | 0.51 |
| Serum triglycerides (mg/dl) | 134 (3) | 120 (4) | 123 (5) | 127 (4) | 155 (6) | 0.69 | 0.15 | <0.0001 | 0.0003 |
| ALT (U/L) | 16.4 (0.3) | 9.0 (0.1) | 12.9 (0.1) | 16.8 (0.1) | 23.5 (0.2) | <0.0001 | <0.0001 | <0.0001 | <0.0001 |
| AST (U/L) | 20.6 (0.2) | 17.3 (0.3) | 19.2 (0.2) | 20.4 (0.2) | 24.0 (0.4) | <0.0001 | <0.0001 | <0.0001 | <0.0001 |
| GGT (U/L) | 25.8 (0.5) | 21.2 (0.8) | 21.8 (0.7) | 25.8 (0.7) | 31.5 (1.1) | 0.46 | <0.0001 | <0.0001 | <0.0001 |
| LDH (U/L) | 153.9 (2.0) | 140.7 (2.1) | 150.4 (2.5) | 155.4 (2.1) | 163.6 (2.4) | 0.0001 | <0.0001 | <0.0001 | <0.0001 |
| Serum Albumin (g/dl) | 4.33 (0.02) | 4.30 (0.03) | 4.34 (0.03) | 4.31 (0.03) | 4.35 (0.03) | 0.17 | 0.52 | 0.0764 | 0.299 |
| Total Bilirubin (mg/dl) | 0.70 (0.01) | 0.67 (0.02) | 0.70 (0.02) | 0.72 (0.02) | 0.70 (0.02) | 0.33 | 0.077 | 0.501 | 0.3 |
| Serum ferritin (ng/mL) | 157 (4) | 135 (5) | 155 (5) | 155 (6) | 170 (8) | 0.016 | 0.0245 | 0.0019 | 0.011 |
| Serum C-reactive protein (mg/dL) | 0.33 (0.01) | 0.38 (0.02) | 0.35 (0.04) | 0.28 (0.01) | 0.32 (0.02) | 0.49 | 0.002 | 0.067 | 0.0052 |

(*Continued*)

**Table 1.** (Continued)

| | Overall | Quartile 1 (<12 IU/L) | Quartile 2 (12–15 IU/L) | Quartile 3 (15–20 IU/L) | Quartile 4 (≥20 IU/L) | P-value* (1 vs. 2) | P-value* (1 vs. 3) | P-value* (1 vs. 4) | Overall# P-value |
|---|---|---|---|---|---|---|---|---|---|
| Serum Creatinine (mg/dl) | 1.18 (0.01) | 1.19 (0.01) | 1.18 (0.01) | 1.17 (0.01) | 1.16 (0.01) | 0.72 | 0.33 | 0.14 | 0.36 |
| Hemoglobin (g/dl) | 15.1 (0.1) | 14.8 (0.1) | 15.0 (0.1) | 15.1 (0.1) | 15.3 (0.1) | 0.0147 | <0.0001 | <0.0001 | <0.0001 |
| Platelets (per cmm) | 257 (2) | 260 (4) | 256 (4) | 253 (3) | 258 (3) | 0.47 | 0.2 | 0.6 | 0.59 |
| Thyroid-Stimulating Hormone (mIU/mL) | 1.75 (0.07) | 1.80 (0.15) | 1.85 (0.30) | 1.71 (0.06) | 1.70 (0.06) | 0.88 | 0.54 | 0.51 | 0.88 |
| **Comorbidities** | | | | | | | | | |
| Diabetes (%) | 222 (5.1%) | 64 (5.5%) | 42 (4.3%) | 65 (5.9%) | 51 (4.7%) | | | | 0.78 |
| Impaired Glucose Tolerance (%) | 1,198 (34%) | 302 (37%) | 248 (30%) | 338 (34%) | 310 (35%) | | | | 0.43 |
| Hypertension (%) | 549 (18%) | 140 (22%) | 129 (20%) | 146 (18%) | 134 (13%) | | | | 0.038 |
| Metabolic Syndrome (%) | 608 (20%) | 139 (20%) | 126 (18%) | 159 (18%) | 184 (23%) | | | | 0.31 |
| HOMA-IR (%) | 587 (17%) | 105 (11%) | 108 (13%) | 160 (17%) | 214 (23%) | | | | 0.0011 |
| Albuminuria (%) | 392 (12%) | 128 (16%) | 85 (13%) | 80 (9%) | 99 (12%) | | | | 0.033 |
| History of MI (%) | 120 (2.9%) | 52 (6.3%) | 27 (3.6%) | 23 (1.9%) | 18 (1.3%) | | | | <0.0001 |
| History of Stroke (%) | 44 (1.3%) | 15 (2.1%) | 9 (1.5%) | 12 (1.1%) | 8 (0.9%) | | | | <0.0001 |
| History of CHF (%) | 86 (1.7%) | 31 (2.9%) | 24 (2.2%) | 17 (1.4%) | 14 (0.8%) | | | | <0.0001 |

Continuous variables presented as mean (standard deviation); categorical variables presented as Number (weighted %). Abbreviations: HOMA-IR = Homeostatic Model Assessment of Insulin Resistance, MI = Myocardial Infarction, CHF = Congestive Heart Failure.

#Overall p-value calculated using Rao-Scott design-adjusted Chi-square test for categorical variables and one-way ANOVA for continuous variables.

*For continuous variables, post-hoc pairwise Tukey comparison tests were conducted. Adjustment for multiple comparisons was not done as comparisons were explorative and did not guide our choice of covariates to include in the analysis.

In both males and females, similar patterns of risk of all-cause mortality were observed when restricting all-cause mortality to death within 10 years (Tables 2–4).

## Modeling the association between ALT quartile and cause-specific mortality

In males, although the unadjusted risks of cardiovascular mortality (HR [95% CI]: 0.54 [0.35, 0.85]) and cancer mortality (0.22 [0.14, 0.36]) were lower in Q4 vs. Q1, this association did not hold after adjusting for covariates. In particular, ALT quartile's association with cardiovascular mortality became insignificant after adjusting for demographic factors (age, race/ethnicity), and ALT quartile's association with cancer mortality became insignificant after adjustment for other liver enzymes. Females in the 3rd quartile had significantly lower hazards of cardiovascular mortality even after adjustment for covariates (aHR [95% CI]: 0.48 [0.28, 0.83]).

Among females with normal BMI, those in Q4 had a significant 68% decreased adjusted risk of cardiovascular mortality relative to those in Q1 (aHR [95% CI]: 0.32 [0.13, 0.83]). This association was not observed in the high BMI group. In males, when stratifying by BMI, we observe a stronger protective association between ALT quartile and cancer mortality in those

**Table 2. Association between ALT quartiles and all-cause and cause-specific mortality in males.**

| | | | Model 1 | Model 2 | Model 3 | Model 4 |
|---|---|---|---|---|---|---|
| | Deaths (%)[#]: 940 (33%) | | HR (95% CI) | HR (95% CI) | HR (95% CI) | HR (95% CI) |
| **All-Cause Mortality** | Quartile 1[¥] | 320 (36%) | 1.00 (Ref) | 1.00 (Ref) | 1.00 (Ref) | 1.00 (Ref) |
| | Quartile 2 | 220 (30%) | 0.79 (0.63, 1.00) | 0.84 (0.64, 1.10) | 0.86 (0.67, 1.10) | 0.85 (0.66, 1.10) |
| | Quartile 3 | 221 (20%) | 0.51 (0.39, 0.66) | 0.61 (0.49, 0.78) | 0.59 (0.47, 0.74) | 0.60 (0.48, 0.74) |
| | Quartile 4 | 179 (15%) | 0.38 (0.28, 0.51) | 0.63 (0.48, 0.82) | 0.58 (0.45, 0.76) | 0.56 (0.42, 0.74) |
| | Deaths (%): 309 (11%) | | | | | |
| **10-Year All-Cause Mortality** | Quartile 1 | 135 (15%) | 1.00 (Ref) | 1.00 (Ref) | 1.00 (Ref) | 1.00 (Ref) |
| | Quartile 2 | 65 (8.2%) | 0.52 (0.31, 0.89) | 0.64 (0.38, 1.08) | 0.70 (0.43, 1.16) | 0.70 (0.43, 1.15) |
| | Quartile 3 | 55 (4.9%) | 0.30 (0.18, 0.52) | 0.47 (0.29, 0.77) | 0.50 (0.30, 0.84) | 0.49 (0.30, 0.79) |
| | Quartile 4 | 54 (4.1%) | 0.25 (0.14, 0.47) | 0.54 (0.30, 0.99) | 0.59 (0.32, 1.07) | 0.48 (0.27, 0.85) |
| | Deaths (%): 249 (6.3%) | | | | | |
| **Cardiovascular Mortality** | Quartile 1 | 84 (8.7%) | 1.00 (Ref) | 1.00 (Ref) | 1.00 (Ref) | 1.00 (Ref) |
| | Quartile 2 | 63 (8.1%) | 0.88 (0.57, 1.37) | 1.00 (0.61, 1.66) | 1.08 (0.67, 1.76) | 1.01 (0.61, 1.68) |
| | Quartile 3 | 52 (4.4%) | 0.46 (0.27, 0.78) | 0.63 (0.37, 1.09) | 0.64 (0.36, 1.16) | 0.60 (0.32, 1.12) |
| | Quartile 4 | 50 (5.2%) | 0.54 (0.35, 0.85) | 1.11 (0.72, 1.69) | 1.08 (0.68, 1.72) | 0.83 (0.46, 1.48) |
| | Deaths (%): 214 (6.1%) | | | | | |
| **Cancer-related Mortality** | Quartile 1 | 90 (11%) | 1.00 (Ref) | 1.00 (Ref) | 1.00 (Ref) | 1.00 (Ref) |
| | Quartile 2 | 48 (6.6%) | 0.54 (0.33, 0.87) | 0.57 (0.34, 0.94) | 0.61 (0.37, 1.02) | 0.70 (0.43, 1.14) |
| | Quartile 3 | 44 (5.3%) | 0.42 (0.26, 0.69) | 0.51 (0.32, 0.81) | 0.54 (0.33, 0.89) | 0.69 (0.45, 1.06) |
| | Quartile 4 | 32 (2.8%) | 0.22 (0.14, 0.36) | 0.35 (0.22, 0.58) | 0.39 (0.22, 0.68) | 0.58 (0.32, 1.06) |

Abbreviations: HR = Hazard Ratio, CI = Confidence Interval.

[¥] -Quartiles for males: $<12$, 12–15, 15–20, $\geq 20$ IU/L

[#]Deaths represented as number of deaths (weighted % of sample)

Model 1 = unadjusted model

Model 2 adjusted for demographic (age, poverty-income ratio, race/ethnicity) and sociobehavioral covariates (alcohol, smoking status)

Model 3 adjusted for Model 2 covariates + cardiometabolic covariates (waist circumference, HDL, systolic BP, triglycerides, C-reactive protein, albuminuria, history of CVD condition)

Model 4 adjusted for Model 3 covariates + liver function-related covariates (albumin, platelet count, AST, total bilirubin)

with normal BMI. Among normal BMI participants, males in Q4 had an 84% lower adjusted risk of cancer mortality relative to those in Q1 (0.16 [0.05, 0.53]).

## Age-related changes in ALT

To better understand the age-related distribution of ALT, we plotted 5-year average ALT levels by sex (S1–S2 Figs). In males, we observed an increase in ALT levels until the 4th decade; there was a linear decrease in ALT thereafter. In females, the pattern is not as clear; ALT levels remain relatively stable. We see a slight increase in AST in females, possibly suggesting sub-clinical liver dysfunction with increasing age, especially after age 50. Upon stratifying analyses by age ($\geq 50$, $<50$), we found that risk of mortality was more pronounced in older male and female individuals, but not significantly different from younger individuals (Table 5).

## Sensitivity analysis

As an additional sensitivity analysis, we determined the dose-dependent association between ALT levels and mortality, treating ALT as a continuous variable (S2 and S3 Tables). After adjusting for covariates, males and females had 3% and 6%, respectively, significantly

**Table 3. Mortality risk for males and females in the highest ALT quartile stratified by BMI Group.**

| | BMI < 25 kg/m$^2$ | | | BMI ≥ 25 kg/m$^2$ | | |
|---|---|---|---|---|---|---|
| | Deaths# (%) | Model 1 (95% CI) | Model 4 (95% CI) | Deaths# (%) | Model 1 (95% CI) | Model 4 (95% CI) |
| | Male | | | | | |
| **All-Cause Mortality** | 386 (30%) | 0.31 (0.18, 0.55) | 0.36 (0.20, 0.64) | 554 (36%) | 0.38 (0.25, 0.57) | 0.77 (0.49, 1.22) |
| **10-Year All-Cause Mortality** | 142 (11%) | 0.23 (0.09, 0.58) | 0.31 (0.12, 0.81) | 167 (11%) | 0.25 (0.12, 0.49) | 0.62 (0.32, 1.21) |
| **Cardiovascular Mortality** | 90 (5.2%) | 0.67 (0.21, 2.16) | 0.53 (0.18, 1.61) | 159 (7.3%) | 0.40 (0.22, 0.74) | 1.19 (0.54, 2.63) |
| **Cancer Mortality** | 89 (5.3%) | 0.06 (0.02, 0.18) | 0.16 (0.05, 0.53) | 125 (6.8%) | 0.33 (0.18, 0.61) | 1.08 (0.40, 2.95) |
| | Female | | | | | |
| **All-Cause Mortality** | 312 (18.8) | 1.04 (0.68, 1.59) | 0.58 (0.36, 0.93) | 549 (26%) | 0.72 (0.50, 1.01) | 0.49 (0.34, 0.71) |
| **10-Year All-Cause Mortality** | 90 (5.4%) | 1.01 (0.50, 2.02) | 0.40 (0.16, 1.00) | 153 (7.3%) | 0.44 (0.24, 0.79) | 0.36 (0.17, 0.79) |
| **Cardiovascular Mortality** | 69 (4.2%) | 0.90 (0.44, 1.86) | 0.32 (0.13, 0.83) | 165 (7.8%) | 0.76 (0.42, 1.39) | 0.55 (0.23, 1.31) |
| **Cancer Mortality** | 93 (5.6%) | 0.63 (0.29, 1.38) | 0.47 (0.17, 1.29) | 127 (6%) | 0.60 (0.28, 1.29) | 0.72 (0.32, 1.62) |

Abbreviations: HR = Hazard Ratio, CI = Confidence Interval.

#Deaths represented as number of deaths (weighted % of sample)

All Hazard Ratios are of the Highest Quartile (Q4; ≥14 IU/L for females and ≥20 IU/L for males) vs. Lowest Quartile (Q1; <9 IU/L for females and <12 IU/L for males)

Model 1 = unadjusted model

Model 4 = fully adjusted model, adjusting for demographic, sociobehavioral, cardiometabolic, and liver function-related covariates

decreased risk of all-cause mortality for every 1 IU/L increase in ALT. This association was greater in magnitude for 10-year all-cause mortality but did not hold for cause-specific mortality.

## Discussion

In this large, nationally representative prospective study, increasing ALT activity within the normal range was associated with decreased risk of all-cause mortality in both males and females. In males, this association was strengthened in those with BMI <25, whereas in females the association was similar across BMI. We also observed significantly decreased risk of cancer mortality in males in the highest quartile of ALT with normal BMI, but not over-weight BMI. In females, whereas overall there was a U-shaped association between ALT quartile and cardiovascular mortality, those with normal BMI had an inverse, dose-dependent association.

The association between ALT and mortality is complex, inconsistent across literature, and affected by several confounding variables [1, 4, 5, 13, 16–24]. Although several studies have identified a potential U-shaped relationship between ALT levels and mortality [7, 9, 10, 13, 16–20], few observational studies have been conducted in subjects with ALT levels within the normal range [5, 13]. Ruhl et al. showed, in a similar NHANES cohort, that serum ALT levels in the 3 lowest deciles were associated with 30–50% increased risk of all-cause mortality in older adults [13]. However, they did not restrict their study population to normal ALT levels or rigorously exclude those with liver disease. Hyeon et al. showed that, within the normal range, those with high-normal ALT levels >20 U/L had increased liver disease mortality compared to those with low-normal ALT levels [5]. However, this study also had a wider range of normal ALT (<40 U/L) that may have included subclinical liver dysfunction. Other studies have reported inverse associations of ALT and mortality in elderly or select populations (e.g. coronary artery disease patients) [18, 21], but there has been inconsistency with exclusion of liver disease. This makes it difficult to compare studies and their magnitudes of association, leading to uncertainty regarding the nature of ALT's association with mortality.

**Table 4. Association between ALT quartiles and all-cause and cause-specific mortality in females.**

| | | | Model 1 | Model 2 | Model 3 | Model 4 |
|---|---|---|---|---|---|---|
| | Deaths (%)[#]: 861 (23%) | | HR (95% CI) | HR (95% CI) | HR (95% CI) | HR (95% CI) |
| **All-Cause Mortality** | Quartile 1[¥] | 193 (19.7%) | 1.00 (Ref) | 1.00 (Ref) | 1.00 (Ref) | 1.00 (Ref) |
| | Quartile 2 | 196 (18.9%) | 0.97 (0.70, 1.34) | 0.88 (0.63, 1.25) | 0.89 (0.63, 1.28) | 0.87 (0.62, 1.23) |
| | Quartile 3 | 215 (18.1%) | 0.92 (0.72, 1.17) | 0.81 (0.65, 1.02) | 0.75 (0.60, 0.95) | 0.69 (0.54, 0.89) |
| | Quartile 4 | 257 (18.2%) | 0.93 (0.68, 1.25) | 0.73 (0.55, 0.98) | 0.64 (0.48, 086) | 0.55 (0.40, 0.77) |
| | Deaths (%): 243 (6.5%) | | | | | |
| **10-Year All-Cause Mortality** | Quartile 1 | 55 (6.0%) | 1.00 (Ref) | 1.00 (Ref) | 1.00 (Ref) | 1.00 (Ref) |
| | Quartile 2 | 58 (4.8%) | 0.79 (0.46, 1.36) | 0.72 (0.43, 1.23) | 0.72 (0.41, 1.25) | 0.70 (0.40, 1.22) |
| | Quartile 3 | 57 (4.0%) | 0.66 (0.39, 1.11) | 0.64 (0.38, 1.08) | 0.59 (0.34, 1.04) | 0.54 (0.30, 0.96) |
| | Quartile 4 | 73 (4.1%) | 0.68 (0.42, 1.11) | 0.59 (0.37, 0.95) | 0.52 (0.31, 0.85) | 0.41 (0.23, 0.72) |
| | Deaths (%): 234 (6.2%) | | | | | |
| **Cardiovascular Mortality** | Quartile 1 | 54 (5.6%) | 1.00 (Ref) | 1.00 (Ref) | 1.00 (Ref) | 1.00 (Ref) |
| | Quartile 2 | 50 (5.3%) | 0.96 (0.61, 1.51) | 0.87 (0.53, 1.42) | 0.82 (0.50, 1.34) | 0.81 (0.46, 1.41) |
| | Quartile 3 | 55 (3.8%) | 0.67 (0.41, 1.09) | 0.60 (0.39,0.92) | 0.49 (0.31, 0.78) | 0.48 (0.28, 0.83) |
| | Quartile 4 | 75 (5.4%) | 0.97 (0.60, 1.56) | 0.76 (0.48, 1.22) | 0.57 (0.35, 0.92) | 0.53 (0.27, 1.07) |
| | Deaths (%): 220 (5.8%) | | | | | |
| **Cancer-related Mortality** | Quartile 1 | 51 (5.8%) | 1.00 (Ref) | 1.00 (Ref) | 1.00 (Ref) | 1.00 (Ref) |
| | Quartile 2 | 58 (6.5%) | 1.12 (0.62, 2.02) | 1.03 (0.57, 1.86) | 1.05 (0.58, 1.91) | 1.09 (0.59, 2.00) |
| | Quartile 3 | 49 (4.6%) | 0.79 (0.44, 1.40) | 0.73 (0.41, 1.30) | 0.71 (0.40, 1.28) | 0.75 (0.39, 1.44) |
| | Quartile 4 | 62 (3.7%) | 0.63 (0.36, 1.10) | 0.53 (0.30, 0.92) | 0.52 (0.29, 0.93) | 0.57 (0.28, 1.16) |

Abbreviations: HR = Hazard Ratio, CI = Confidence Interval.

[¥]- **Quartiles for females: <9, 9–11, 11–14, ≥14 IU/L**

[#]Deaths represented as number of deaths (weighted % of sample)

Model 1 = unadjusted model

Model 2 adjusted for demographic (age, poverty-income ratio, race/ethnicity) and sociobehavioral covariates (alcohol, smoking status)

Model 3 adjusted for Model 2 covariates + cardiometabolic covariates (waist circumference, HDL, systolic BP, triglycerides, C-reactive protein, albuminuria, history of CVD condition)

Model 4 adjusted for Model 3 covariates + liver function-related covariates (albumin, platelet count, AST, total bilirubin)

To avoid this biasing effect of ALT as an inflammatory biomarker, we rigorously restricted our ALT ranges to normal levels. We also excluded other major causes of liver disease, such as non-alcoholic fatty liver disease as per ultrasound, excessive alcohol consumption, viral serologies, transferrin saturation, and AST levels.

Although there is no established mechanism for the inverse association between ALT levels and all-cause mortality, we speculate that ALT's role in skeletal muscles and the liver may be contributory. ALT is involved in the glucose-alanine cycle, which creates substrates for gluconeogenesis and protein synthesis—thus, increases in ALT levels within the normal range may correspond to increased liver functionality [3].

Skeletal muscle also affects ALT activity by utilizing the enzyme in the reverse glucose-alanine cycle [10]. Low ALT levels may reflect decreased muscle mass or decreased blood flow to muscle cells. Several studies have shown that decreased ALT in older age groups is associated with frailty and decreased muscle strength [6, 25, 26]. It is also well established that sarcopenia is a risk factor for mortality [27]. Furthermore, in males especially, our results demonstrated that those with normal BMI have pronounced protective effects of ALT compared to those who are overweight or obese. In line with a study by Nakamura et al., this may suggest that, in

**Table 5. Mortality risk for males and females in the highest ALT quartile stratified by age group.**

| | Age ≥ 50 | | | Age < 50 | | |
|---|---|---|---|---|---|---|
| | Deaths[#] (%) | Model 1 (95% CI) | Model 4 (95% CI) | Deaths[#] (%) | Model 1 (95% CI) | Model 4 (95% CI) |
| | Male | | | | | |
| **All-Cause Mortality** | 731 (66%) | 0.41 (0.28, 0.58) | 0.55 (0.39, 0.77) | 209 (12%) | 1.15 (0.61, 2.15) | 0.81 (0.39, 1.69) |
| **10-Year All-Cause Mortality** | 245 (22%) | 0.26 (0.14, 0.50) | 0.45 (0.21, 0.97) | 64 (4%) | 0.91 (0.30, 2.74) | 0.94 (0.27, 3.72) |
| **Cardiovascular Mortality** | 208 (19%) | 0.59 (0.33, 1.06) | 0.83 (0.41, 1.71) | 41 (2.6%) | 1.76 (0.59, 5.25) | 0.86 (0.30, 2.48) |
| **Cancer Mortality** | 176 (16%) | 0.32 (0.16, 0.61) | 0.67 (0.32, 1.40) | 38 (2.2%) | 0.40 (0.09, 1.72) | 0.62 (0.06, 5.94) |
| | Female | | | | | |
| **All-Cause Mortality** | 665 (55%) | 0.66 (0.48, 0.89) | 0.56 (0.39, 0.81) | 196 (8%) | 0.53 (0.26, 1.09) | 0.46 (0.17, 1.21) |
| **10-Year All-Cause Mortality** | 183 (15%) | 0.51 (0.30, 0.87) | 0.38 (0.18, 0.80) | 60 (2.5%) | 0.47 (0.18, 1.22) | 0.45 (0.21, 0.97) |
| **Cardiovascular Mortality** | 205 (17%) | 0.67 (0.42, 1.07) | 0.58 (0.27, 1.27) | 29 (1.2%) | 0.47 (0.09, 2.41) | 0.26 (0.02, 4.28) |
| **Cancer Mortality** | 139 (12%) | 0.48 (0.26, 0.91) | 0.48 (0.21, 1.11) | 81 (3.2%) | 0.51 (0.20, 1.30) | 0.65 (0.20, 2.19) |

Abbreviations: HR = Hazard Ratio, CI = Confidence Interval.

[#]Deaths represented as number of deaths (weighted % of sample)

All Hazard Ratios are of the Highest Quartile (Q4; ≥14 IU/L for females and ≥20 IU/L for males) vs. Lowest Quartile (Q1; <9 IU/L for females and <12 IU/L for males)

Model 1 = unadjusted model

Model 4 = fully adjusted model, adjusting for demographic, sociobehavioral, cardiometabolic, and liver function-related covariates

the absence of excess adiposity, muscle mass and subsequent increases in ALT may be protective against all-cause mortality [23].

While the primary purpose of this study was not to compare risk of mortality between males and females, we observed potential age-related differences in ALT's association with mortality. When plotting mean ALT levels in 5-year intervals, following a peak in the 4th decade, ALT decreases linearly in males, which is consistent with existing literature [28, 29]; in females, no such obvious relationship existed, although Dong et al. does report decreasing ALT in females as well [28]. Lean muscle mass follows a similar course, declining rapidly in the 4-5th decade, more so in males than females [30, 31]. Females experience a rapid decrease in muscle mass approximately a decade after menopause, likely the result of a loss in estrogenic protection [31].

There were also potential sex differences in the association between ALT quartile and cardiovascular mortality. Whereas in males, there was no association, in females, there was a U-shaped association. This U-shaped association may be due to lack of exclusion of all inflammation-related ALT elevations in females as noted by the significant increase in metabolic syndrome in the third and fourth ALT quartiles and the rise, albeit insignificant, in C-reactive protein (in males, metabolic syndrome was similar across quartiles and C-reactive protein decreased with increasing ALT).

Males and females with normal BMI, but not elevated BMI, had an inverse association between ALT quartile and cancer mortality. Although there was no statistical difference between BMI groups, point estimates for those with normal BMI were more protective, possibly suggesting presence of adipose tissue modifies the association between ALT (and relatedly, lean muscle mass) and cancer mortality. Alternatively, elevated BMI may suggest subclinical fatty liver disease that was not detected through ALT elevations.

There were several limitations in this study. Firstly, baseline characteristics and ALT levels were measured cross-sectionally only at the initial examination. ALT levels and several covariates are known to fluctuate and could have changed over our long follow-up period. However, such non-differential misclassification bias would likely have underestimated our association, rather than overestimate it. Secondly, given that ALT activity is stable for only three days in

proper storage, there is a possibility that the ALT levels may have been underestimated due to logistical issues when they were measured. Thirdly, although we aimed to exclude any liver dysfunction and other confounding, there may have been residual confounding that we did not account for in the analysis. Additionally, this data was from 1988–1994; population demographics have changed considerably in the United States since then (e.g. Hispanic groups other than Mexican Americans have become more prevalent, as well as Asian ethnicities) and thus results may not be nationally representative of the population today. Lastly, we used the National Death Index to determine mortality status, which can be subject to misclassification with regards to cause of death.

Despite these limitations, there were also several strengths. To our knowledge, this is the first study of the association between ALT and mortality with a follow-up period of over 20 years, allowing us to observe the long-term effects of ALT on mortality. Furthermore, we applied strict and comprehensive inclusion and exclusion criteria to prevent confounding and to create a study population without any liver disease, including subclinical liver dysfunction. Additionally, we used a large, nationally representative study cohort with significant power, allowing for generalizability of our results to the US population.

## Conclusion

In conclusion, ALT levels were inversely associated with all-cause mortality in both male and females, with greater reduction in risk in males $\geq$50 or BMI <25. The relationship between ALT and cause-specific mortality is inconclusive. Serum ALT is an inexpensive and widely available assay, and it has significant potential as a tool to help clinicians evaluate and manage patients with multiple metabolic comorbidities such as diabetes mellitus, coronary artery disease, and other components of the metabolic syndrome. Future research should focus on understanding the clinical relevance of serum ALT as a prognostic indicator.

## Supporting information

**S1 Table. Baseline characteristics by ALT quartile in females.**
(DOCX)

**S2 Table. Association between ALT as a continuous variable and all-cause and cause-specific mortality in males.**
(DOCX)

**S3 Table. Association between ALT as a continuous variable and all-cause and cause-specific mortality in females.**
(DOCX)

**S1 Fig. Mean ALT, AST, and LDH in males stratified by 5-year age groups.** Error bars indicate 1 standard deviation. This is a cross-sectional display of mean liver enzyme values across age groups, not changes in individual values with time.
(DOCX)

**S2 Fig. Mean ALT, AST, and LDH in females stratified by 5-year age groups.** Error bars indicate 1 standard deviation. This is a cross-sectional display of mean liver enzyme values across age groups, not changes in individual values with time.
(DOCX)

## Author Contributions

**Conceptualization:** Aayush Visaria, Suraj Pai.

**Data curation:** Aayush Visaria.

**Formal analysis:** Aayush Visaria, Suraj Pai.

**Investigation:** Aayush Visaria.

**Methodology:** Aayush Visaria, Neil Kothari.

**Software:** Aayush Visaria.

**Supervision:** Aayush Visaria, Alla Fayngersh, Neil Kothari.

**Writing – original draft:** Aayush Visaria, Suraj Pai.

**Writing – review & editing:** Aayush Visaria, Suraj Pai, Alla Fayngersh, Neil Kothari.

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
