## [Decision Letter · Decision Letter 0]

3 Nov 2020

Association between alanine aminotransferase within the normal range and all-cause and cause-specific mortality: A nationwide cohort study

PONE-D-20-27592

Dear Dr. Visaria,

We’re pleased to inform you that your manuscript has been judged scientifically suitable for publication and will be formally accepted for publication once it meets all outstanding technical requirements.

Kind regards,

Emre Bozkurt

Academic Editor

PLOS ONE

Journal Requirements:

1. Your ethics statement should only appear in the Methods section of your manuscript. If your ethics statement is written in any section besides the Methods, please move it to the Methods section and delete it from any other section. Please ensure that your ethics statement is included in your manuscript, as the ethics statement entered into the online submission form will not be published alongside your manuscript.

Additional Editor Comments (optional):

Congragulations to you for this nice article. Your article is appropriate to be accepted by considering the positive reviewer comments.

Reviewers' comments:

Reviewer's Responses to Questions

**Comments to the Author**

1. Is the manuscript technically sound, and do the data support the conclusions?

Reviewer #1: Yes

Reviewer #2: Yes

2. Has the statistical analysis been performed appropriately and rigorously? 

Reviewer #1: Yes

Reviewer #2: Yes

3. Have the authors made all data underlying the findings in their manuscript fully available?

Reviewer #1: Yes

Reviewer #2: Yes

4. Is the manuscript presented in an intelligible fashion and written in standard English?

Reviewer #1: Yes

Reviewer #2: Yes

5. Review Comments to the Author

Reviewer #1: dear author

I evaluated your work titled 'Association between alanine aminotransferase within the normal range and all-cause and cause-specific mortality: A nationwide cohort study'. Although it is a well-known marker ALT study, it is a scientifically interesting study.

No comment

Reviewer #2: The paper is well designed and the issue is interesting.

Methodology of the study is well described. Limitations and biases were clearly explained and detailed.

References are powerful and up to date.

Good paper.

6. PLOS authors have the option to publish the peer review history of their article (what does this mean?). If published, this will include your full peer review and any attached files.

Reviewer #1: No

Reviewer #2: No

---

## [Editor Report · Acceptance letter]

6 Nov 2020

PONE-D-20-27592 

Association between alanine aminotransferase within the normal range and all-cause and cause-specific mortality: A nationwide cohort study 

Dear Dr. Visaria:

I'm pleased to inform you that your manuscript has been deemed suitable for publication in PLOS ONE. Congratulations! Your manuscript is now with our production department. 

Kind regards, 

on behalf of

Dr. Emre Bozkurt 

Academic Editor

PLOS ONE